# Assessment of the Overtourism Phenomenon Risk in Tunisia in Relation to the Tourism Area Life Cycle Concept

**Monika Widz \* and Teresa Brzezińska-Wójcik**

Department of Regional Geography and Tourism, Faculty of Earth Sciences and Spatial Management, Maria Curie-Sklodowska University, al. Kraśnicka 2d, 20-031 Lublin, Poland; tbrzezin@poczta.umcs.lublin.pl
\* Correspondence: monika.widz@umcs.pl; Tel.: +48-81-537-6852

**Abstract:** Tunisia is a destination where organised mass tourism has prevailed since 1985. This trend is still being observed, despite the unstable geopolitical situation in North Africa. Current reports from booking portals indicate that this country will be one of the most popular tourist destinations in 2020. Therefore, the aim of the study was to determine the prospects for sustainable development in Tunisia in 2020–2025 as means to prevent the negative effects of overtourism. The research was conducted in three stages: (1) analysis of the phases of tourism development in Tunisia from 1960 to 2019 in relation to the Tourism Area Life Cycle concept, (2) identification of the destination's evolution in 2015–2019 with the method of trend function exploration, and (3) an attempt to assess the risk of overtourism in Tunisia in light of Tourism Carrying Capacity on the basis of the Tourism Intensity Index and Tourism Density index. The study results revealed three phases of development in Tunisia, i.e. exploration, involvement, and development. The verification of the trend function indicated that Tunisia would enter the consolidation phase in 2020. The highest risk of overtourism is estimated for three governorates—Tunis, Sousse, and Monastir.

**Keywords:** tourism area life cycle model; tourism carrying capacity; tourism intensity index; tourism density index; overtourism

## 1. Introduction

Tunisia, which is regarded as a "3S" destination, has been one of the most popular destinations for years chosen by international tourists due to its diverse tourist offerings. Tourism services and investments not only generate jobs, but also influence regional and local development.

In every area, tourism, like most types of economic activity, is associated with measurable profits and losses that should be assessed. An important element is the available area and other features of the region as well as their response to tourist traffic. When "the impact of tourism on a destination or parts thereof excessively influences perceived quality of life of citizens and/or quality of visitors' experiences in a negative way", the phenomenon can be referred to as overtourism [1]. As formulated by Peeters et al. [2], "overtourism describes a situation in which the impact of tourism, at certain times and in certain locations, exceeds physical, ecological, social, economic, psychological, and/or political capacity thresholds". As emphasised by Dodds and Butler [3], this is a new term for an old problem, i.e. the presence of an excessive number of tourists in a certain area, which may exert a negative impact on the place. The literature on this issue has been reviewed by Dodds and Butler [4], Milano et al. [5] and Kruczek [6].

The phenomenon of overtourism has been documented primarily in urban areas (e.g., Kraków, Poland—Kruczek [7]) and in protected areas, especially in national parks (e.g., Cinque Terre National

Park, Italy—Faccini et al. [8]), on coasts (e.g., Maya Bay—Phi Phi Leh, Thailand—Dodds [9]), on entire islands (e.g., Mallorca, Spain—Garcia and Servera [10]), and in rural areas (e.g., Bled, Slovenia—Mihalič et al. [11]). However, the problem has been assessed with different methods so far, including qualitative, e.g., Koens et al. [12], and quantitative methods based on the tourism carrying capacity model, e.g., Bertocchi et al. [13]. A review study of overtourism in as many as 41 countries (the selection was based on a set of criteria including one case per EU country, an even distribution over the four types of destinations—rural, urban, coastal and islands, heritage and attractions, and 12 iconic non-EU destinations) was conducted by Peeters et al. [2], who proposed a conceptual model of this phenomenon. The importance of studies conducted by Manera and Valle [14] who carried out a comparative analysis of the overtourism phenomenon in all countries of the world, should be emphasised as well.

The overtourism phenomenon in an area requires an immediate response from entities involved in tourism. This activity is usually strongly associated with the concept of sustainable tourism. It ensures the use of geographical environment resources in such a way that future generations may benefit from them. Therefore, overcrowding of the tourist areas or damage to their natural, cultural, recreational, and specialist assets should be prevented. It is essential that the touristic area does not become a tourist slum or lose its tourist function completely [15]. It is therefore important to determine the current stage of tourism development in a destination and to plan its further development.

The tourism planning process is gradual; it is characterised by continuity and comprehensiveness and focuses on achievement of sustainable development [16]. Its goal is to generate income and jobs on one hand and to protect resources ensuring tourist satisfaction on the other [17]. The space planning process for the needs of tourism development in compliance with the concept of sustainable development is becoming increasingly important in view of long-term prospect and attempts to maintain harmony between the natural environment, local community, and economic development. It is highly important that the Tourism Carrying Capacity should be defined in this process [2,14,18]. Excessive concentration of tourist traffic in a given area not only leads to overcrowding in attractive sites, but also reduces the level of aesthetic sensations experienced by tourists and results in natural environment degradation [7].

The concept of carrying capacity in tourism originates in the 1960s [19,20], although the problem (carrying capacity) was mentioned for the first time in 1936 [21]. Many definitions of the tourism carrying capacity (TCC) have been proposed e.g., [22], but the most comprehensive one has been formulated by the World Tourism Organization [23], i.e. tourism carrying capacity is "the maximum number of people that may visit a tourist destination at the same time, without causing destruction of the physical, economic, socio-cultural environment and an unacceptable decrease in the quality of visitors' satisfaction". The definition of the tourism carrying capacity concept gave rise to the search for a method to calculate an indicator [14] that facilitates determination of threshold values as the key factor for sustainability and preserve environment's good condition [24].

Studies on this problem in Tunisia (e.g., [25,26]) emphasised the need to plan sustainable tourism development in this country. This is associated with the documented problems in sustenance/preservation of natural resources, especially in the coastal zone (e.g., [27,28]). However, the phenomenon of overtourism was not documented. One of the recent studies highlighting this phenomenon [14] has demonstrated that Tunisia belongs to the group of countries with a moderate value of the tourism intensity index (422—mean value calculated for 1995–2015). This is important, as many reports based on data from booking portals [29,30] show that this country will be one of the most popular tourist destinations in 2020. The number of bookings made in this country by British people alone has increased by as much as 96.1% [29].

The main aim of the study is to assess the risk of overtourism in Tunisia based on the allowable volume of tourist traffic. The assessment is supported by analysis of three specific issues: (1) phases of tourism development in relation to the tourism area life cycle model with identification of the current tourism phase in this country (2) determination of the tourism carrying capacity for each

governorate with using selected indicators, and (3) identification of governorates that are most exposed to overtourism risk and possibilities to prevent this phenomenon.

## 2. Materials and Methods

The assessment of the risk of overtourism in the analysed area is a complex issue. Therefore, it was divided into three basic stages: diagnosis, prognosis, and assessment of the overtourism risk (Figure 1).

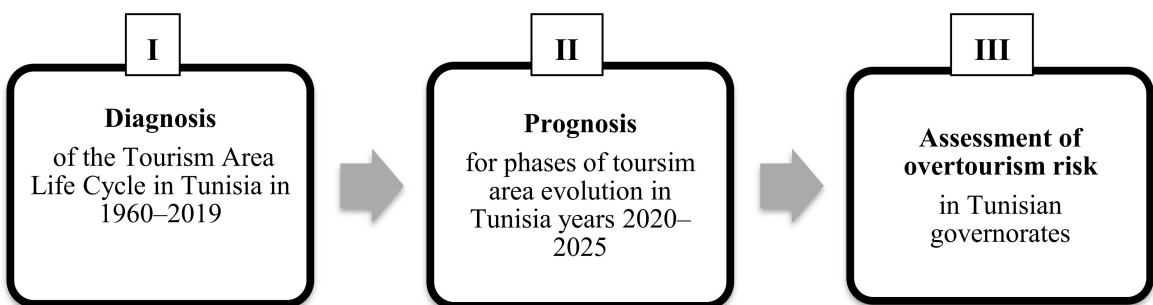

**Figure 1.** Scheme of the research design. Source: own study.

The first stage of the study (Figure 1) consisted of data analysis and phases identification for tourism development in Tunisia vs. the concept of the tourism area life cycle (TALC) [31], taking into account the diagnosis of the tourist area definition proposed by Alejziak [32]. The TALC model assumes that there are seven phases of tourist area evolution: exploration, involvement, development, consolidation, stagnation, decline, or rejuvenation. Although it originates from the concept of the tourist product cycle, the model can be suitable for analysis of a "3S" destination, such as Tunisia, as a recreational area (in the subtropical and tropical climate zone). As distinguished by Rak, Pstrocka-Rak [33], this destination is included in the group of two-generation tourist reception areas (the first generation—tourists arriving by rail during the tourist centers heyday in the 19th century, the second one—tourists arriving by plane in the second half of the 20th century). The TALC model facilitates determination of the relationship of an increase in the number of tourists and hotel beds with the number of residents over a specified time [34]. The identification of the development phases in Tunisia as a tourist reception area using the TALC model in 1960–2019 included the number of international tourists, and the number of hotel beds.

The second stage of the research (Figure 1) consisted of an attempt to determine the evolution phase in Tunisia in 2020–2025 using the trend function exploration method. The method facilitates the prediction of the tourist traffic volume should trends towards development and unintended fluctuations in time series occur [35]. This stage was focused on establishment of a hypothetical but most likely scenario for the future tourist traffic.

The choice of the analytical model, i.e. the trend function (linear, logarithmic, power, exponential, and polynomial) was based on calculated values of the determination coefficient (R-squared) for 2015–2019 for the following variables: the number of international tourists, the number of hotel beds, and the number of residents. The coefficient shows how some changes in the dependent variable are explained by changes in the explanatory variable. This facilitates assessing which of the analysed models is well fitted [36] (p. 209). The prognosis for the evolution of Tunisia as a tourist destination was based on a trend line analysis.

The third stage of the study (Figure 1) was an attempt to assess the risk of overtourism in the governorates of Tunisia. In the literature, the pre-evaluation stage consists of the tourism carrying capacity determination [2,14], which is usually measured with the two most frequently used indicators—the tourism intensity index (TII) and tourism density (TD) (Figure 2).

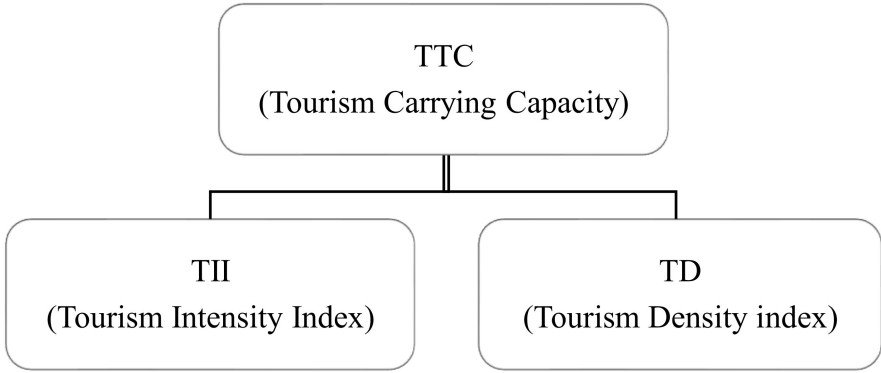

**Figure 2.** Tourism Carrying Capacity model in view of overtourism. Source: Own analysis based on Manera and Valle [14], Peeters et al. [2].

Different variables are used for these indicators in the literature. For example, Manera and Valle [14] used the following variables to calculate TII: the number of tourists, population, tourism revenue, gross domestic product (GDP) for a specific country, and GDP for the world, whereas TD in a given country was defined as the number of tourists per $km^2$. In turn, Peeters et al. [2] used two variables, i.e. bed-nights and inhabitants, for calculation of TII and bed-nights per $km^2$ for TD. In the present study, the definitions and classification of TII and TD follow those proposed by Peeters et al. [2]. The variables proposed by these authors are best suited to the analysis of Tunisian administrative units.

The tourism intensity index (TII) is the ratio of nights spent at tourist accommodation establishments relative to the total resident population of the area:

$$\frac{bed-nights}{inhabitants} \text{ TII } = \text{ bed} - \text{nights/inhabitants}$$

The Tourism Density index (TD) is defined as the annual number of bed-nights per $km^2$:

$$\frac{bed-nights}{km^2} \text{TD } = \text{ bed} - \text{nights/km}^2$$

Next, for governorates classified at the high and highest risk of the overtourism phenomenon according to the TII and TD indicators (Table 1), we formulated recommendations for the further use of their tourist potential and the possibility to divert tourist traffic to the lowest risk or low risk areas.

**Table 1.** Classification of the overtourism risk based on the tourism intensity index and tourism density index.

| Indicator Values | | Degree of Risk | Interpretation of Risk |
|---|---|---|---|
| Tourism Intensity Index | Tourism Density | | |
| <407 | <3.18 | 1 | Lowest risk |
| 407.1–719 | 3.18–4.49 | 2 | Low risk |
| 719.1–1174 | 4.50–6.30 | 3 | Medium risk |
| 1174.1–2278 | 6.31–9.58 | 4 | High |
| >2278 | >9.58 | 5 | Highest risk |

Source: Own analysis based on Peeters et al. [2], Eurostat [37], Peeters [38], World Bank Group [39].

The study involves secondary data sources usage. In the first stage (preparation of the TALC model), data from 1960–2019 were analysed. The figures were provided by two sources: (1) the Institut National de la Statistique (INS) [40]—the "number of hotel beds" variable and (2) the Office National du

Turisme Tunisien (ONTT) [41]—the "number of international tourists" variable (information obtained in person).

The second stage of the study (prognosis for the next phase of Tunisia evolution as a tourist area) was based on two data sources as well: 1) the Office National du Turisme Tunisien [41]—the "number of international tourists in 2005–2019" variable (information obtained in person) and the Institut National de la Statistique [40,42,43]—the "number of hotel beds and number of residents" variables.

The third stage of the study (assessment of the overtourism risk phenomenon in Tunisian governorates based on the tourism intensity index and tourism density) was conducted using 2018 data from all governorates. They were obtained from the reports of the Institut National de la Statistiques [42,43] and Commissariat Général de Développement Régionale (CGDR) [44]. These were data on the area, number of residents, and number of bed-nights in each governorate.

## 3. Results

### 3.1. Evolution of Tunisia as A Tourist Reception Area in 1960–2019

Three phases of Tunisia development, i.e. exploration, involvement, and development, were identified from the curve constructed in accordance with the TALC model proposed by Butler [31] (Figure 3).

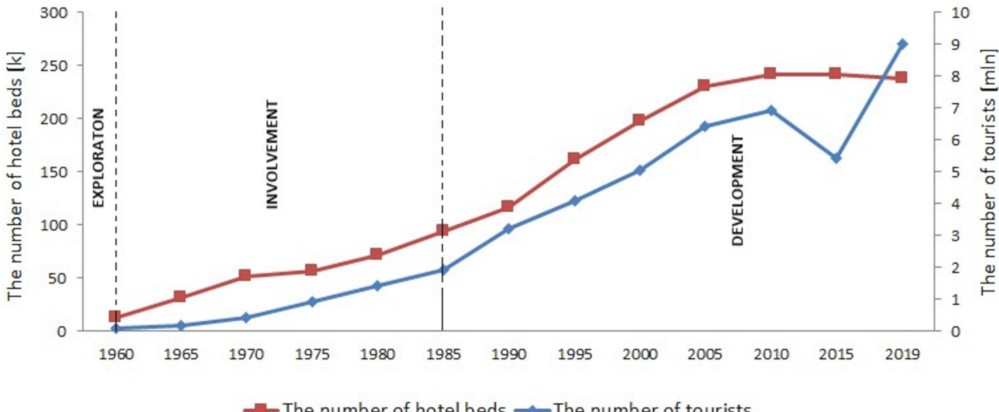

**Figure 3.** Tunisia evolution phases as a tourist reception area in 1960–2019. Source: Own analysis of ONTT data.

The first phase, exploration, lasted in Tunisia until the 1960s, i.e. before the appearance of stable tourism investments. This period was characterised by a relatively small number of the explorer-type tourists ranging from several thousand to 94,000 in 1960 (Figure 3). This type of tourists chose their travel destination based on the desire to meet local people and explore natural and cultural resources, to experience adventure, and to rest in a remote isolated and undiscovered area, which was not affected by the lack of a tourist base [33]. At this phase of area development, the presence of tourists exerts a relatively low impact on the economic and social life of residents [45].

The involvement phase (1960–1985) was characterised by a substantial increase in the number of tourists in Tunisia, i.e. 1.8 million in 1985, which was almost 20-times higher than in 1965. During this period, a 10-year tourism development plan for the country was implemented to provide 35,000 hotel beds. At that time, the first tourist areas were established near coastal towns, e.g., Tunis, Hammamet, Nabeul, Sousse, Monastir, and Djerba [46]. Additionally, the Office National du Turisme Tunisien (ONTT) was established to develop and promote tourism and to ensure high quality of tourism services. The Société Hôtelière et Touristique de Tunisie (SHTT) was created to supervise and manage the construction of national hotels. Consequently, 15,000 workplaces were created in the tourism industry [47].

　　　The mass-scale development of tourism noted after 1985 initiated the development phase in Tunisia (Figure 3). In this phase, the number of hotel beds was increased to over 93,000 in 1985 and 40,000 jobs were created in the tourism sector [48]. Over the next 10 years (1990–2000), the number of hotel beds doubled (197,500) as well as the number of jobs (79,000) [49]. The development of the hotel base and economic changes, especially in 1987 [46], contributed to an increase in the tourists' number. In 1990, it reached over 3 million visitors and increased significantly to approximately 5 million tourists within the next 10 years (1990–2000) [49].

　　　A noticeable increase in tourist traffic was reported at the turn of the 20th and 21st centuries. This generated increasing tourism-related income and development of infrastructure. Despite the global crisis and the concern about H1N1 influenza worldwide, an increase in the number of tourist arrivals was noted in Tunisia, especially in 2008 (approx. 7 million) (Figure 4).

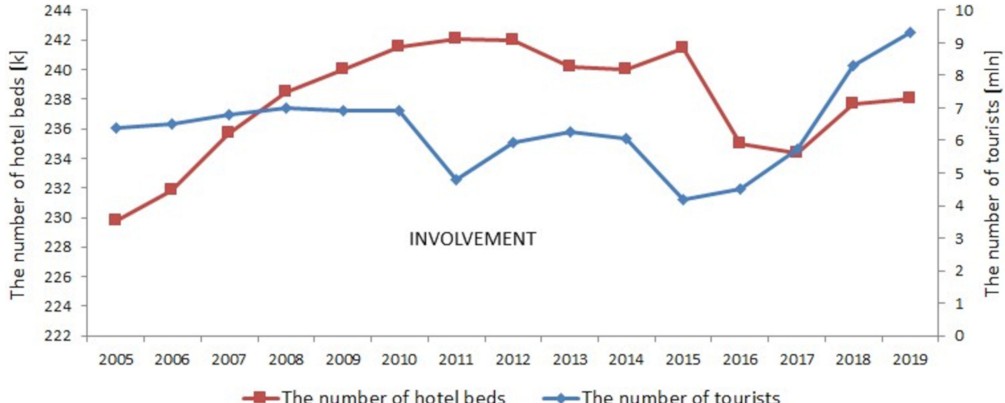

**Figure 4.** Tunisia evolution phase as a tourist reception area in 2005–2019. Source: Own analysis of ONTT data.

　　　In 2011–2015, there were two sharp declines in the number of visitors to Tunisia (Figure 4). The first was reported during the first three months of 2011 when the number of tourist arrivals decreased by as much as 44% compared to 2010, and the tourism industry losses amounted to over 620 million USD. Another decrease was noted in August 2015 when only 4.2 million tourists came, which was 24% less than in the corresponding period of 2014, and the number of tourists from Europe decreased by 50% [47]. The decline in the tourist traffic resulted in a reduction of the number of hotels beds from 242,100 to 240,000 in 2013 and from 241,400 in 2015 to 235,000 in 2016 (Figure 4).

　　　In 2018, the number of tourists increased again to reach 8.3 million, and the number of hotel beds increased to 237.600 (Figure 4). Even higher tourist traffic (9.4 million visitors) was recorded in Tunisia in 2019; it accounted for a 13.6% increase compared to 2018. This was mainly associated with the greater number of tourists coming from European countries—an increase by almost 2.8 million (15.9%) and from North African countries—by almost 5 million (15.5%) [50].

*3.2. Prognosis for the Next Phase of Tunisia Evolution as A Tourist Area*

　　　The verification of the trend function models for the 2015–2019 time series (Table 2) at the level of the coefficient of determination R-squared demonstrated the highest values of the polynomial function for the analysed variables—number of tourists ($R^2 = 0.963$), number of residents ($R^2 = 0.958$), and number of hotel beds ($R^2 = 0.751$).

　　　Therefore, the prognosis for the future phase of Tunisia evolution (2020–2025) was based on the 2nd order polynomial trend line. Given the values of the determination coefficients close to 1.0, it can be concluded that the function accurately describes the development trend for the analysed variables.

**Table 2.** Coefficients of determination R-squared in 2015–2019.

| Trend Function | Values of Coefficients of Determination R-Squared for Variables | | |
| --- | --- | --- | --- |
| | Number of Tourists | Number of Residents | Number of Hotel Beds |
| Linear | 0.938 | 0.882 | 0.055 |
| Logarithmic | 0.809 | 0.727 | 0.193 |
| Power | 0.854 | 0.730 | 0.190 |
| Exponential | 0.955 | 0.884 | 0.053 |
| Polynomial | 0.963 | 0.958 | 0.751 |

Source: Own analysis of ONTT data.

In the analysed time series, an upward development trend was noted for all variables, i.e. the number of hotel beds, tourists, and residents, starting in 2015. The intersection of the trend lines for the "number of residents" and "number of tourists" variables will be observed in 2020 (Figure 5). This may indicate that Tunisia is entering the next phase of the tourism area evolution referred to as consolidation. It is characterised by an equal or higher number of tourists than the number of residents [31].

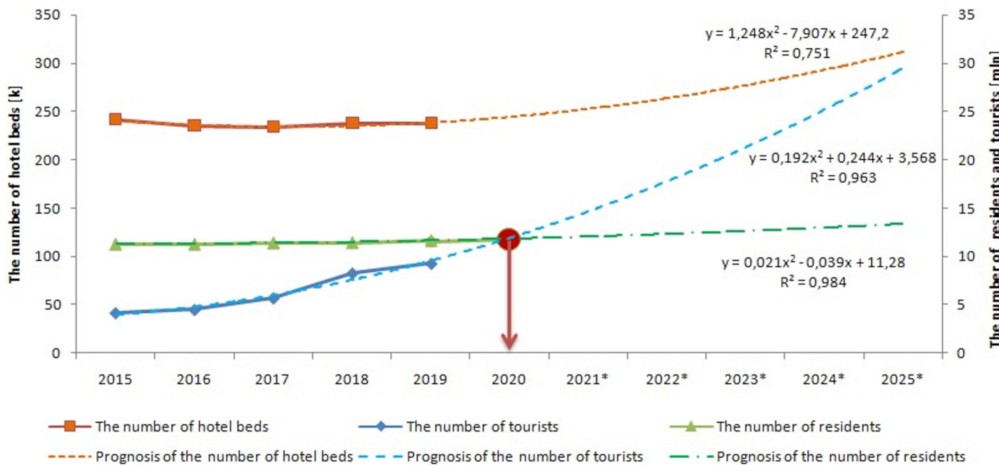

**Figure 5.** Prognosis for the next phase of Tunisia evolution in time series 2015–2025. Source: Own analysis of ONTT data.

*3.3. Assessment of the Overtourism Risk Phenomenon in Tunisian Governorates Based on the Tourism Intensity Index and Tourism Density*

The prognosis for the phase of Tunisia evolution as a tourist area in 2015–2019 determined by the trend function models verification for the time series was the basis for identification of the tourist carrying capacity and potential overtourism in all governorates.

Given the criteria for classification of the overtourism risk based on the value of the Tourism Intensity Index (Table 1), it should be underlined that the highest risk (5) was demonstrated for three out of the 24 Tunisian governorates analysed, i.e. Tunis, Sousse, and Monastir. Some governorates were assigned the following levels of risk: Nabeul—high risk (4), Medenine—medium risk (3), and Mahdia—low risk (2). The other 18 governorates were identified as the lowest-risk areas (Figure 6).

As shown by the criteria for classification of the overtourism risk based on the value of the Tourism Density index (Table 2), the highest risk (5) was identified in two governorates—Medenine and Sousse. Two administrative units, Monastir and Nabeul, were assigned medium risk (3) and low risk (2) was detected in one governorate, i.e. Mahdia. As indicated by the TD values, the other 19 governorates should be classified in the category of the lowest risk (Figure 7).

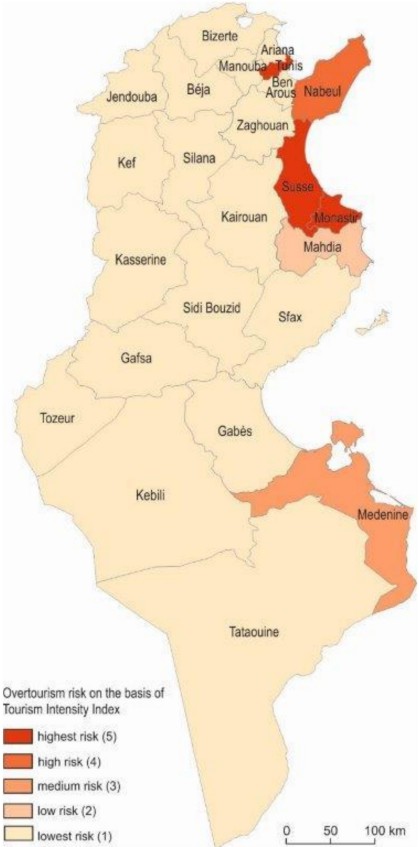

**Figure 6.** Overtourism risk in Tunisia based on the tourism intensity index. Source: own study.

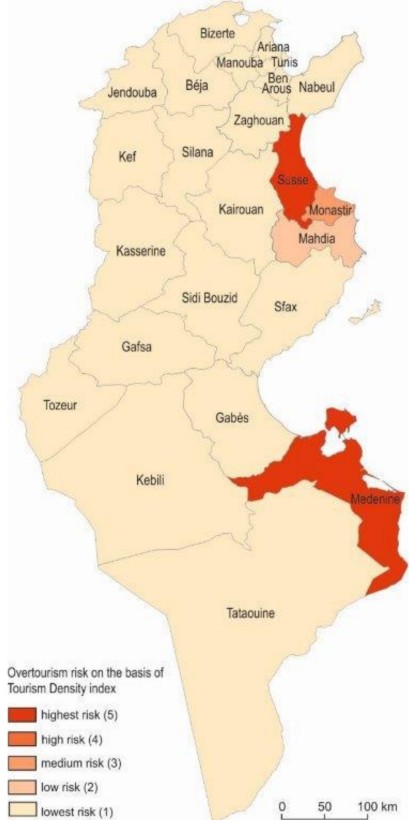

**Figure 7.** Overtourism risk in Tunisia based on the tourism density index. Source: own study.

## 4. Discussion

The interpretation of the TALC curve obtained for the 1960–2019 data from Tunisia (Figure 3) raises no doubts as to the first two phases—exploration (before 1960) and involvement (1960–1985). In contrast, the development phase, especially in 2010–2019, may seem disputable. The reported reduction in the number of tourists and hotel beds in 2011 and 2015 followed by an increase in these parameters should not be interpreted as the decline and rejuvenation phases. Firstly, Tunisia has not yet reached either the consolidation or the stagnation phases that precede the decline phase. Secondly, the decrease in the tourist traffic did not result from deterioration of the services' quality or attractiveness of the offer. As emphasised by Zmyślony [51], the cause of the "developmental anomalies" in both cases was the strong impact of internal developmental determinants.

The cause of the sharp decline in the tourist traffic in 2011 was the "Jasmine Revolution", which induced social and economic changes in the country and abroad. Consequently, the changing geopolitical situation brought serious repercussions on the tourist markets of almost all North African and Middle East countries. This resulted in a change in destinations and structure of tourist traffic. In Tunisia, this led to cancellation of most of the tourist events booked for the 2011 season, although the revolution lasted less than a month (from 17 December 2010 to 14 January 2011). The collapse of the tourist traffic in 2015 was caused by the terrorist attacks at the Bardo National Museum in Tunis (18 March 2015) and on the beach in Sousse (26 June 2015). Therefore, one should agree with the suggestion made by Mika [52] that all manifestations of social and political tensions are immediately reflected by reduction of the tourist traffic volume in countries affected by such unrest. Thus, the geopolitical stability of Tunisia as a tourist region and tourists' safety are of key importance for the modern tourist market development.

The development phase implies that the tourist market is well defined. As emphasised by some authors [53,54], residents in this phase lose control over the development of the tourism market and local services are replaced by external investors associated mainly with hotel services. Additionally, the local authorities involvement in the development of tourism declines, as the design and area development takes place at the regional and national levels. Resident antagonisms may also be sometimes observed.

The next stage in the development of Tunisia may be the consolidation phase. Its characteristic feature is an increasing number of tourists, who may exceed the number of residents over time [41]. The involvement of residents proceeds through joining the business activity for tourists or focusing services mainly or even exclusively on visitors [53].

The subjective selection of the 2015–2019 time series for the prediction process was prompted by the visual assessment of the graph (Figure 4). It presented an upward development trend and random fluctuations of the analysed variables. The reduction of the hotel beds number and tourists recorded in 2011 and 2015 was caused by factors unrelated to tourism phenomena. As in the case of the TALC curve for 1960–2019, the fluctuations may have affected the trend function analysis and exerted a negative effect on the interpretation. Therefore, it was reasonable to carry out the prediction process for 2020–2025 based on the trend function models for the time series 2015–2019, which reflects the current and real dynamics of tourism in Tunisia. It revealed that Tunisia would enter the *consolidation* phase in 2020. Furthermore, the number of tourists may be twice as high as the number of residents in 2025 (Figure 5), which may lead to overtourism in this area. Therefore, it is important at this stage to take appropriate actions in governorates threatened with the greatest risk of this phenomenon.

As demonstrated by the tourism intensity index and tourism density, the phenomenon of overtourism threatens the governorates of Sousse (the highest risk), Monastir, and Madanin. All these areas are located on the east coast of Tunisia on the Mediterranean Sea. Currently, the Sousse and Madanin governorates specialise in long-term seven-night stays, which are typical of only 20% of the area of Tunisia.

What attracts tourists to these governorates? In addition to its recreational values, Sousse (also called the "Pearl of the Sahel") has unique cultural assets, e.g., the Medina of Sousse (kasbah, ramparts,

medina, the Great Mosque, Bu Ftata Mosque, ribat), included on the UNESCO World Heritage List [55]. This area has an extensive tourist infrastructure, especially in Port El Kantaoui, where a yacht port, golf courses, and numerous hotels can be found [56].

The Madanin Governorate covers a geographically diverse area. Tourist traffic is typically concentrated on Djerba Island and Matmata-Dahar Plateau. Djerba Island, located in the Gulf of Gabes, is an important recreational area attracting tourists with its sandy beaches and numerous thalassotherapy centers. A great attraction on the Matmata-Dahar Plateau, famous for its "bad land" loess landscape [57] known as "lunar" landscape, is the culture of Berber descendants [58] called troglodytes. The "lunar landscape" near Matmata was the location for some scenes in the "Star Wars" movie series [56]. Monastir attracts tourists mainly with the Monastir-Skanes seaside resort with luxury hotels along sandy beaches [59,60].

It should be emphasised that, although they visit attractions or do sightseeing, most tourists stay in the coastal zone in the three governorates. The tourist traffic overload of the coastal area (overtourism) and the related problems (water stress, excess waste, shore line developments), especially in Hammamet Bay and on Djerba island, are part of research focused on the deteriorating quality of the natural environment (e.g., [28]) and the negative impact of tourism on the natural environment (preservation of biodiversity and the natural character of the coast, fresh water supplies for tourists, wastewater treatment, electricity consumption) and society (e.g., [26,27]).

## 5. Conclusions

Using the TALC model, three development phases were identified in Tunisia – exploration (before 1960), involvement (1960–1985), and development (1985–2019).

The prognosis carried out using the trend function models indicates that the next phase of Tunisia evolution as a tourism area will be the consolidation phase. It will begin in 2020. The number of tourists will increase significantly and gradually exceed the number of residents. As predicted, the number of tourists in 2025 may be twice as big as the number of residents, which implies the risk of overtourism in the analyzed area. The consolidation stage was a premise to take measures to investigate the degree of overtourism in this destination.

The increasing tourists' number strengthens the tourism function of the area and has a positive effect on economic development at the regional level on the one hand, but can exert a negative impact on the natural environment and, consequently, worsen the recreational conditions on the other hand. It can also lead to the buildup of social tensions at various levels.

At present, however, the values of the tourism intensity index and tourism density, which reflect the tourism carrying capacity, indicate the risk of overtourism in three governorates: the highest in Sousse (TII 5, TD 5) and high in Monastir (TII 5, TD 3) and Madanin (TII 3, TD5).

The research procedure adopted in the present study yields a diagnosis of the overtourism problem in the analysed area. However, it brings preliminary results, which diagnose the problem based on a limited number of variables and represent entire governorate areas. In future, the phenomenon of overtourism should be analysed taking into account a greater number of variables, e.g., the TCC phenomenon. It is also important to diagnose overtourism in basic administrative units within the governorates, which may be difficult due to the limited possibility of acquiring relevant data.

In compliance with the principles of sustainable development, specific measures should be undertaken to prevent the phenomenon of overtourism in Sousse, Monastir, and Madanin and, in the near future, also in Tunis and Nabeul. Some of them are highlighted by Halioui and Schmidt [26], UNWTO [1], and Kruczek [6]. Actions that may limit the negative effects of overtourism should include the following steps: 1. creation of new attractions and tourist routes in neighboring governorates in order to disperse visitors in space and time; 2. adjustment of legal provisions for organization and management of tourism in the governorates; 3. providing the local community with the benefits from tourism.

**Author Contributions:** Conceptualization, M.W.; methodology, M.W. and T.B.-W.; validation, M.W.; formal analysis, M.W. and T.B.-W.; investigation, M.W. and T.B.-W.; resources, M.W. and T.B.-W.; data curation, M.W.; writing—original draft preparation, M.W. and T.B.-W.; writing—review and editing, M.W. and T.B.-W.; visualization, M.W.; supervision, T.B.-W. All authors have read and agree to the published version of the manuscript.

**Funding:** This research received no external funding.

**Conflicts of Interest:** The authors declare no conflict of interest.

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
