# Peer review of "Assessment of the Overtourism Phenomenon Risk in Tunisia in Relation to the Tourism Area Life Cycle Concept"

_sustainability, doi:10.3390/su12052004_

Round 1

Reviewer 1 Report

The manuscript is methodologically correct, logically structured and well-written (cohesive paragraphs, articulate statements). The English is very good no misprints were detected.

The introduction provides a good, generalized background of the topic and appropriate data analyses have been conducted. Results and discussion are balanced and well supported. The reference list seems exhaustive.

My only remark concerns the source of data used for the analyses. Authors refer to booking portals (in the abstract and introduction) for the tourism trend in Tunisia in 2020, but it is not clear if these sources were also used for calculation of the indices and/or additional reference were considered (national/regional reports, etc.). Authors should explain that better.

Author Response

Dear Reviewer

Thank you very much for the positive opinion and valuable remarks. Attached, you will find our comments on the review.

Response to Reviewer 1 Comments

Point 1: My only remark concerns the source of data used for the analyses. Authors refer to booking portals (in the abstract and introduction) for the tourism trend in Tunisia in 2020, but it is not clear if these sources were also used for calculation of the indices and/or additional reference were considered (national/regional reports, etc.). Authors should explain that better.

Response 1: The presentation of data provided by booking portals in both the Abstract and the Introduction was intended to show the current increase in the interest in Tunisia as a tourist destination. The data refer to forecasts on tourism in 2020 for the entire country.

The indicators for the individual governorates were calculated based on data shown in national and regional reports, with no reference to forecasted data mentioned above. The relevant comment made by the Reviewer prompted the authors to include an additional paragraph on the source of data in the final part of section 2. Materials and Methods (lines 191-206). It presents the sources of data used in the research.

The study involves secondary data sources usage. In the first stage (preparation of the TALC model), data from 1960–2019 were analysed. The figures were provided by two sources: 1) the Institut National de la Statistique (INS) [43] – the “number of hotel beds” variable and 2) the Office National du Turisme Tunisien (ONTT) [44] – the “number of international tourists” variable (information obtained in person).

The second stage of the study (prognosis for the next phase of Tunisia evolution as a tourist area) was based on two data sources as well: 1) the Office National du Turisme Tunisien [44] – the “number of international tourists in 2005–2019” variable (information obtained in person) and the Institut National de la Statistique [43,45,46] – the “number of hotel beds and number of residents” variables.

The third stage of the study (assessment of the overtourism risk phenomenon in Tunisian governorates based on the Tourism Intensity Index and Tourism Density) was conducted using 2018 data from all governorates. They were obtained from the reports of the Institut National de la Statistiques [45,46] and Commissariat Général de Développement Régionale (CGDR) [47]. These were data on the area, number of residents, and number of bed-nights in each governorate.

Therefore, the reference to data sources in lines 136-137 has been deleted. These data are referred to at the end of section 2.

Yours faithfully,

The Authors

Reviewer 2 Report

In my opinion, I feel that study of the overtourism phenomenon risk is a relevant  topic and would provide an excellent contribution to the growing literaturein this area. While the data collected for this study has the promise to help in this capacity, the introduction/lit review and the methodology as stated by the authors is a fatal flaw. The limited scope misses some rather pertinent and impactful literature and contributions in this space that would significantly improve the breadth of the manuscript. Others main concerns is that the methodology is not clear explained and the discussion  is quite narrow to the point that worthwhile research was missed in this important section. The conclusions are scarce.

This manuscript would benefit from a thorough proofread and edit.

Author Response

Dear Reviewer

Thank you very much for your positive opinion and valuable remarks. Attached, you will find our comments on the review.

Response to Reviewer 2

Point 1: The limited scope misses some rather pertinent and impactful literature and contributions in this space that would significantly improve the breadth of the manuscript.

Response 1: We agree with the comment. The literature on overtourism is very extensive. Especially in the last two years, several monographic papers and many articles focused on this issue have been published. The following relevant references have been added:

  1. Dodds, R.; Butler, R. The phenomena of overtourism: a review. International Journal of Tourism Cities 2019, 5, 519–528.
  2. Dodds, R.; Butler, R. (Eds.) Overtourism: Issues, realities and solutions; De Gruyter: Berlin, Boston, Germany, USA, 2019.
  3. Milano, C.; Cheer, J.M.; Novelli, M. (Eds.) Overtourism: Excesses, Discontents and Measures in Travel and Tourism. CABI: Wallingford, Oxfordshire and Boston, MA, 2019
  4. Koens, K., Postma, A., Papp, B. Is overtourism overused? Understanding the impact of tourism in a city context. Sustainability 2018, 10, 4384.
  5. Bertocchi, D.; Camatti, N.; Giove, S.; van der Borg J. Venice and Overtourism: Simulating Sustainable Development Scenarios through a Tourism Carrying Capacity Model. Sustainability 2020, 12, 512.
  6. Coccossis, H.; Mexa, A. (Eds.) The challenge of tourism carrying capacity assessment: Theory and practice. Aldershot: Ashgate Publishing Ltd., UK, 2004.
  7. Pstrocka, M. Issues Concerning Tourist Carrying Capacity in the English Language Literature. Turyzm/Tourism 2004, 14, 91–103.
  8. Manning, R.E. 2002. How Much is Too Much? Carrying Capacity of National Parks and Protected Areas. In Proceedings International Conference on Monitoring and Management of Visitor Flows in Recreational and Protected Areas, Arnberger, A., Brandenburg, C., Muhar, A. Eds.; Bodenkultur University Vienna: Vienna, Austria, 2002; pp. 306–313. Available online: http://mmv.boku.ac.at/ downloads/mmv1-proceedings.pdf (accessed on 16 02 2020).
  9. Saarinen, J. Traditions of sustainability in tourism studies. Annals of Tourism Research 2006, 33, 1121–1140.
  10. Chapoutot, J.M.M. Profil de durabilité dans quelques destinations touristiques méditerranéenes – La destination Jerba en Tunisie. Plan Bleu, Sophia Antipolis, 2011.
  11. Agoubi, B., Kharroubi, A., Abida, H. Hydrochemistry of groundwater and its assessment for irrigation purpose in coastal Jeffara aquifer, southeastern Tunisia. Arabian Journal of Geosciences 2013, 6, 1163–1172.

Point 2: Others main concerns is that the methodology is not clear explained and the discussion is quite narrow to the point that worthwhile research was missed in this important section.

Response 2: We agree with the comment and therefore we have added information and explanations in section 2 on Materials and Methods (164-172 and 191-206). The discussion part has been extended as well (lines 382-389).

Point 3: The conclusions are scarce.

Response 3: The conclusions have been extended in response to the comment (lines 398-404 and 409-415).

Point 4: This manuscript would benefit from a thorough proofread and edit.

Response 4: Following this comment, the authors have had the text proofread and edited by an English translator. 

Yours faithfully,
The Authors

Reviewer 3 Report

This work focuses on a very interesting topic: the overtourism in Tunisia. The theoretical framework grounds the problem under study through a complete and current range of references. The objectives are correctly defined. The methodology is well-designed and is consistent with the objectives of the study. The interpretation and discussion of results is clear, objective and consistent. The conclusions summarize well the results obtained and are consistent with the work presented. However, there are some aspects that can be improved.

In lines 122-123, Figure 5 is referred to without having previously referred to Figures 2, 3 and 4. All elements should be referred to sequentially, so it is suggested that this aspect be co-corrected.

In lines 136 and 143 the formulas for TII and TD contain "bed-nights". The "-" can be confused with a "minus", so it is suggested to correct this notation.

In lines 200 to 203, the use of "," and “.” in R2 should be standardized.

The legend of Figures 6 and 7 should be revised. It may be clearer to present these figures separately. There is no reference to Figure 6.

Author Response

Dear Reviewer

Thank you very much for the positive opinion and valuable remarks. Attached, you will find our comments on the review.

Point 1: In lines 122-123, Figure 5 is referred to without having previously referred to Figures 2, 3 and 4. All elements should be referred to sequentially, so it is suggested that this aspect be co-corrected.

Response 1: The reference to Figure 5 in lines 122-123 was meant to direct the reader to the results obtained with the method described, i.e. the trend function exploration method. However, we agree with the comment that referring to a figure representing results in the methods' description disrupts the order of referencing to all figures in the text. The reference to Figure 5 has been deleted accordingly (line 151).

Point 2: In lines 136 and 143 the formulas for TII and TD contain "bed-nights". The "-" can be confused with a "minus", so it is suggested to correct this notation.

Response 2: We agree with the comment. Unfortunately, the hyphen “-” in fractional notation, is treated as a “minus” in MS Word, which we overlooked. Therefore, the formula has been changed from fractional to linear (lines 175-176 and 184-185).

Point 3: In lines 200 to 203, the use of "," and “.” in R2 should be standardized.

Response 3: The error in the notation of R2 has been corrected.

Point 4: The legend of Figures 6 and 7 should be revised. It may be clearer to present these figures separately. There is no reference to.

Response 4: We agree with the comment that the legend of Figures 6 and 7 should be revised. We have placed the captions separately (lines 310-313). The position of the figures - next to each other - remains unchanged. In our opinion, this arrangement of the figures facilitates visual comparison between research results for both indicators. Reference to Figure 6 has been included in line 301.

Yours faithfully,
The Authors

Round 2

Reviewer 2 Report

The authors have attended the review comments and they have significantly improved the quality of the paper. 

The authors have attended the review comments and they have significantly improved the quality of the paper.